# Polylactide/Polyvinylalcohol-Based Porous Bioscaffold Loaded with Gentamicin for Wound Dressing Applications

**DOI:** 10.3390/polym13060921

**Published:** 2021-03-17

**Authors:** Maliheh Amini Moghaddam, Antonio Di Martino, Tomáš Šopík, Haojie Fei, Jaroslav Císař, Martina Pummerová, Vladimír Sedlařík

**Affiliations:** Centre of Polymer Systems, University Institute, Tomas Bata University in Zlin, tr. Tomase Bati 5678, 760 01 Zlin, Czech Republic; Amini@utb.cz (M.A.M.); dimartino@utb.cz (A.D.M.); sopik@utb.cz (T.Š.); haojie@utb.cz (H.F.); jcisar@utb.cz (J.C.); pummerova@utb.cz (M.P.)

**Keywords:** Polylactide, porous bioscaffolds, thermal treatment, additives, sustained release, wound dressing

## Abstract

This study explores the feasibility of modifying the surface liquid spraying method to prepare porous bioscaffolds intended for wound dressing applications. For this purpose, gentamicin sulfate was loaded into polylactide-polyvinyl alcohol bioscaffolds as a highly soluble (hygroscopic) model drug for in vitro release study. Moreover, the influence of inorganic salts including NaCl (10 g/L) and KMnO_4_ (0.4 mg/L), and post-thermal treatment (T) (80 °C for 2 min) on the properties of the bioscaffolds were studied. The bioscaffolds were characterized by scanning electron microscopy, Fourier Transform infrared spectroscopy, and differential scanning calorimetry. In addition, other properties including porosity, swelling degree, water vapor transmission rate, entrapment efficiency, and the release of gentamicin sulfate were investigated. Results showed that high concentrations of NaCl (10 g/L) in the aqueous phase led to an increase of around 68% in the initial burst release due to the increase in porosity. In fact, porosity increased from 68.1 ± 1.2 to 94.1 ± 1.5. Moreover, the thermal treatment of the Polylactide-polyvinyl alcohol/NaCl (PLA-PVA/NaCl) bioscaffolds above glass transition temperature (T_g_) reduced the initial burst release by approximately 11% and prolonged the release of the drug. These results suggest that thermal treatment of polymer above T_g_ can be an efficient approach for a sustained release.

## 1. Introduction

Wound dressings have improved vastly over the past decade, from crude, traditional gauze to tissue-engineered scaffolds. An extensive variety of wound dressing formats based on their application for the wound have been investigated or made commercially available [1,2]. A desirable wound dressing should provide an optimal healing environment, which leads to rapid wound healing. It should be able to absorb exudates from the wound surface, maintain a proper moist environment in the wound bed, and, crucially, it should be able to gradually release antimicrobial active agents to ensure prolonged antimicrobial activity and sustain a healthy concentration of healing tissues [3,4]. In recent years, there has been growing attention to fabricate porous structures for wound dressing applications based on natural and synthetic polymers via different techniques. Polylactide (PLA) is one of the biopolymers that has been widely used in wound dressing applications due to its biocompatibility, biodegradability, good mechanical durability, and non-toxicity to the human body [5,6]. However, PLA is highly hydrophobic, which severely restricts its application in the field of wound dressing due to the limitation of water uptake capacity. Moreover, hydrophobicity of PLA has a great impact on the efficiency of the release of the active agents from the polymer matrix [5,7,8,9]. Therefore, porous structure could be one approach in order to tackle these issues. A three-dimensional porous structure can improve the permeability of the wound dressing, and at the same time provide sufficient space for cell growth. It has previously been reported that porous structures promote skin regeneration and wound healing [5,10]. Furthermore, porous structures offer a larger surface area that could allow active agents to diffuse outwards from the matrix in an extremely efficient manner [9].

PVA is a synthetic biocompatible polymer widely used in biomedical applications [11]. It is an excellent choice for improving the properties of PLA due to its nontoxicity, hydrophilicity, low cost, and ease of processing [12,13]. Ribba et al. fabricated PLA-PVA films for wound dressing applications, which exhibited good stability over time, humidity control, and biocompatibility [14]. Hongyan et al. developed the PLA-PVA/SA membranes and assessed their in vivo and vitro wound healing capability. These membranes displayed an effective wound healing performance [11]. Moreover, the manufacturing of PLA-PVA scaffolds is cost-effective and easy on the large scale. Both PLA and PVA have been approved by the FDA for use in biomedical applications [13,15,16]. However, as mentioned above, tissue repair is difficult in non-porous PLA-PVA scaffolds. Therefore, there is a need to investigate the porosity of PLA-PVA scaffolds for wound dressing applications [11,17].

Surface liquid spraying is a unique and simple method for the preparation of a porous structure (porous bioscaffolds). It is similar to the emulsion freeze-drying technique. However, the only difference is that in the surface liquid spraying method, the organic phase is sprayed to the aqueous phase [18,19]. In this technique, an organic compound is dissolved in an organic solvent, which is then sprayed into an aqueous polymer solution to form an oil-in-water emulsion. The principle of the freeze-drying technique is the sublimation process, in which the frozen water in the polymer is directly converted from solid to gas state without apparent liquefaction [20,21]. Generally, the emulsion freeze-drying technique is the most widely used method due to its relative ease of use. However, this method has two major restrictions for hydrophilic drugs in the hydrophobic polymer matrix: (1) Poor encapsulation efficiency (EE), and (2) fast release of active agents occurring at the early stages [22,23,24,25,26]. Several studies have reported that the addition of salts to the polymer could enhance the efficiency of encapsulation by depressing drug aqueous solubility, while simultaneously depressing organic solvent solubility in the aqueous phase [22,23,24]. Pistel et al. demonstrated that the addition of inorganic additives, such as salts, to the aqueous phase increases porosity and the specific surface area of the polymer [27]. One approach to overcome the drug release in early stages is thermal treatment, which occurs by heating the polymer matrixes above the (T_g)._ Thermal treatment above T_g_ could change the physical-mechanical properties of polymer and therefore prolong the drug release [28,29,30]. Castro et al. reported sustained release of simvastatin from polylactide acid (PLA) membranes after thermal treatment [31].

In this study, PLA-PVA porous bioscaffolds were fabricated by the surface liquid spraying method for potential wound dressing application. Sodium chloride (NaCl) and potassium permanganate (KMnO_4_) salts were added to the aqueous phase of an oil in water (O/W) emulsion. NaCl is a neutral salt while KMnO_4_ is a strong oxidizing agent and has mild antiseptic and astringent properties, which traditionally has been used to treat exuding wounds [32,33]. Finally, after a short thermal treatment (80 °C, 2 min) of the bioscaffolds, the effect of the salts (NaCl, KMnO_4_) and the thermal treatment on the in vitro release profile, entrapment efficiency and physicochemical properties of the bioscaffolds were investigated.

## 2. Materials and Methods

Polylactide semi-crystalline PLA6202D with M_W_ of 97 k Da was purchased from NatureWorks^®^ Ingeo™ (Minnetonka, MN, USA). Chloroform was purchased from VWR (14 Media Village, Leighton Buzzard, LU7 0GA, UK). Gentamicin sulfate salt (G-1264) and polyvinyl alcohol with M_W_ of 47 k Da, degree of hydrolysis: 98.0–98.8, were supplied by Sigma-Aldrich (St. Louis, MO, USA). Sodium chloride was purchased from Mikrochem (Slovak Republic). Potassium permanganate was purchased from Penta Prague (Czech Republic).

### 2.1. Preparation of the Porous Bioscaffolds

The organic phase was 2% (*w*/*v*) PLA in chloroform solution and the aqueous phase was 0.1% (*w*/*v*) PVA aqueous solution. Different salt concentrations were added to the aqueous phase (10 g/L NaCl and 0.4 mg/L KMnO_4_). The resultant organic solution was sprayed onto the PVA aqueous solution at the rate of 4 mL/min under moderate magnetic stirring (600 rpm) and 1 bar air-pressure at room temperature to form an oil in water (O/W) emulsion. Subsequently, stirring was maintained overnight in order to evaporate the chloroform. Next, the product was washed three times with deionized water (DI) and filtered. The loading of GS was performed by dissolving GS in DI water and dispersing the filtered product into the solution of GS and DI water. The final product was then frozen. In the final step, the frozen sample was lyophilized. Figure 1 shows the schematic diagram of the surface liquid spraying process.

### 2.2. Thermal Treatment of the Porous Bioscaffolds

In order to study the effects of thermal treatment, the porous bioscaffolds were placed in an oven (Memmert, Germany) at 80 °C for 2 min. The bioscaffolds were then kept in the silica gel containing desiccator. Subsequently, thermal treated bioscaffolds were then subjected to various characterization tests as described below.

### 2.3. Attenuated Total Reflection-Fourier-Transform Infrared (ATR-FTIR) Spectroscopy

The ATR-FTIR test is an appropriate approach to identify functional groups of polymers and molecular structures of the chemicals. The FT-IR spectra were obtained on a Nicolet iS10 instrument equipped with a Zn/Se crystal (Thermo Fisher Scientific, Waltham, MA). The collected spectra in the wavenumber range from 400 to 4000 cm^−1^ represented the average of 64 scans at a spectral resolution of 4 cm^−1^. The spectra for bioscaffolds before and after the thermal treatment were recorded.

### 2.4. Differential Scanning Calorimetry (DSC)

Thermal properties of bioscaffolds before and after thermal treatment were investigated by DSC on a DSC1 STAR system (Mettler Toledo, Greifensee, Switzerland). The preparation of samples has been adopted from our previous paper [34]. In brief, approximately 5 mg of each bioscaffold was placed in an aluminum pan. The following heating program was applied in the presence of Nitrogen with the flow rate of 50 mL min^−1^: In an initial heating cycle, bioscaffolds were heated from 25 °C to 180 °C (10 °C min^−1^), with the highest temperature maintained for 5 min. This was followed by a subsequent cooling to −35 °C (10 °C min^−1^). The temperature of −35 °C was maintained for 5 min, and a further heating scan was performed at 220 °C (10 °C min^−1^). The melting temperature (T_m_) and exothermal response related to cold crystallization temperature (T_c_) were obtained from the second heating cycle. The region of glass-transition temperature (T_g_) was determined from the second heating scan. The degree of crystallinity (χ_c_) was calculated according to the Equation (1) [34]:(1)$$\chi_{c}=\left(\frac{{\Delta H}_{m}-{\Delta H}_{c}}{{\Delta H}_{m}^{0}}\right)\;\times \;100\%$$
where ΔH_m_ is the heat of fusion, ΔH_c_ represents the enthalpy of cold crystallization, and ${\Delta H}_{m}^{0}$ is the tabulated heat of fusion for a theoretically 100% crystalline PLA homopolymer (93.1 J g^−1^) [34].

### 2.5. Morphological Analysis

The morphology of the bioscaffolds was investigated by thermionic-emission scanning electron microscopy (ESEM) (VEGA-II LMU, TESCAN, Brno, Czech Republic) in order to visualize the porosity of the bioscaffolds. The bioscaffolds were prepared by cryogenic fracturing of the bioscaffolds in liquid nitrogen. This was followed by coating bioscaffolds with a thin layer of Au/Pd. The microscope equipped with an SE detector was operated at an acceleration voltage of 10kV in the high-vacuum mode [34].

### 2.6. Porosity Measurement

The porosity of the bioscaffolds was determined by liquid displacement method [35]. The weight (W_1_) and volume (V) of bioscaffolds were measured before the immersion into ethanol. After the saturation of bioscaffolds by the absorption of ethanol, the weight of bioscaffolds were measured again (W_2_). The porosity of the bioscaffolds was calculated according to the Equation (2) (ρ represents the density of the ethanol) [35]. All experiments were performed in triplicate.
(2)$$\mathrm{Porosity}\left(\%\right)\;=\;\left(\frac{W_{2}-W_{1}}{\rho V}\right)\;\times \;100$$

### 2.7. Swelling Test

The swelling test was performed by immersing pre-weighted samples, which had been cut into 1 cm^2^ square pieces, in phosphate-buffered saline (PBS) solution of pH 7.4 at 37 °C. Subsequently, the samples were removed at specific time intervals and the excess water on the surface was carefully absorbed by using filter paper, after which the samples were weighted. The degree of swelling was calculated according to the following Equation [36].
(3)$$\mathrm{degree}\;\mathrm{of}\;\mathrm{swelling}\;\left(\%\right)\;=\;\left(W_{i}-W_{t}\right)/W_{i}\;\times \;100$$
where W_t_ is the weight of the swollen sample, and W_i_ is the sample at time zero (starting time).

### 2.8. Water Solubility

To determine the water solubility of the PVA fraction presented in bioscaffolds, samples (samples were cut into 1 cm^2^ square pieces) were placed in an oven at 37 °C for 24 h. Then, the dried bioscaffolds were immersed in DI water for 24 h and dried again at 37 °C for 24 h. The solubility of the bioscaffolds was evaluated using the following Equation [37]:(4)$$S=\left(\frac{W_{0}-W_{d}}{W_{0}}\right)\;\times \;100$$
where W_0_ and W_d_ represent the weights of the dry sample and the weight of the dried sample at 37 °C after immersion in distilled water, respectively [37].

### 2.9. Water Vapor Transmission Rate (WVTR)

The moisture permeability of the bioscaffolds was determined by measuring the WVTR according to the standard ASTM test method (E96-90) as follows [38]. The samples were cut into discs with a diameter of 15 mm, and mounted on top of a plastic tube containing 10 mL of distilled water. These assemblies were sealed with Teflon tape in order to avoid boundary loss and then placed in a straight position at 37 °C inside an oven containing 1 kg of freshly dried silica gel, to maintain relatively low humidity conditions. After regular intervals of time, the weights of the assemblies were measured. The WVTR of the samples was calculated according to the following Equation [39]:WVTR (gr/m^2^ · Day) = (W_i_ − W_f_)/A(5)
where A is the exposure area, and W_i_ and W_f_ are the initial and final weights of the assemblies respectively.

### 2.10. Solvent Residue Analysis

Chloroform residue was measured by a headspace autosampler AOC-5000 connected to the GCMS-QP2010 Ultra device (Shimadzu, Kyoto, Japan) equipped with a fused silica capillary column (Stabilwax, 30 m × 0.25 mm × 0.25 µm, Restek, PA, USA). Headspace samples were prepared in 20 mL gas tied vials filled with cca 0.05 mg of sample for an equilibrium time of 30 min at 95 °C. Helium was used as the carrier gas, at a flow rate of 1.26 mL/min. The temperature of the injection was maintained at 230 °C at a split ratio of 1:20, the volume of the injected sample equalling 1 mL. The temperature of the column was held at 50 °C for 1 min, and then increased to 70 °C at a rate of 25 °C/min, and increased to 240 °C for additional 2 min. The temperatures of the ion source (EI, 70 eV) and the interface were set at 200 °C and 240 °C, respectively. The range of the scan was 25–250 (m/z) at event time 0.3. The peaks obtained in the resulting TIC spectrum were identified with the help of the NIST11 Spectra Library.

### 2.11. Drug Loading and In Vitro Release Study

The gentamicin sulfate (GS) loading efficiency and in vitro release behavior of the drug was carried out in the presence of a non-toxic and isotonic release medium, PBS. The entrapment efficiency (EE) and loading capacity (LC) of GS were evaluated by immersing the bioscaffolds into 50 mL of PBS10 mM at pH 7.4 in capped glass flasks. The glass flasks were kept in an orbital incubator (Stuart SI500, UK) at 37 ± 0.5 °C, set to 40 rpm for 1 h. Equations (6) and (7) were applied to calculate EE (%) and LC (%), respectively [40].
(6)$$\mathrm{EE}\left(\%\right)=\frac{{\mathrm{Total}}_{\mathrm{GS}}-{\mathrm{Free}}_{\mathrm{GS}}}{{\mathrm{Total}}_{\mathrm{GS}}}\;\times \;100$$
(7)$$\mathrm{LC}\left(\%\right)=\frac{{\mathrm{Total}}_{\mathrm{GS}}-{\mathrm{Free}}_{\mathrm{GS}}}{\mathrm{matsweight}}\;\times \;100$$
where Total_GS_ was the amount of primary GS added to the solution, and Free_GS_ was assessed through high-performance liquid chromatography (HPLC), on a Dionex UltiMate 3000 Series device (Thermo Fisher Scientific, Germany) using o-phtaldialdehyde (OPA) as a derivatizing agent. The methodology was adapted from Smělá et al., [41] with minor modifications, as described below. The reducing solution was prepared by adding 250 µL of 2-mercaptoethanol and 10 mL of 0.04 M sodium borate (pH 11). Separately, a solution of OPA was prepared by dissolving 2.5 g of OPA in a mixture composed of 400 µL methanol, 200 µL reducing solution and 4.4 mL of 0.04 M sodium borate (pH 11). The OPA solution was stored in the dark and utilized within 24 h after preparation. The separation after OPA in needle derivatization was performed on a reversed-phase column XSELECT CSH C18 5 µm (4.6 × 250 mm; Waters, USA), equipped with a security guard column (Phenomenex, USA) at 30 °C. A mixture of 100 mM Acetate buffer (A; pH 5.8) and HPLC grade Acetonitrile (ACN; B) was applied as the mobile phase (55:45, *v/v*) at a flow rate of 0.4 mL/min. The volume of the injection was defined by user defined program (UDP) settings, a volume of 10 µL was injected into the column originating from a drawn sample volume of 1 µL. The eluted OPA derivatives were detected by fluorescence using 330 nm and 440 nm as excitation and emission wavelengths, respectively. All measurements were performed in triplicate.

In the next stage, to evaluate the in vitro release rate, the medium was replaced with 50 mL fresh PBS 10mM (pH 7.4), and glass flasks were kept in an orbital incubator at 37 ± 0.5 °C, 40 rpm. At predefined time intervals, 2 mL of the medium was taken, and the same volume of the fresh medium was replaced in the glass flasks. The amount of GS in the medium was evaluated by HPLC apparatus, and OPA was used as a derivatizing agent [41]. The cumulative percentage release of GS from the bioscaffolds was calculated and plotted against time (*n* = 3).

### 2.12. Statistical Analysis

All experiments were done in triplicates, and the data were presented as mean ± standard deviation. One-way ANOVA analysis was carried out using the Graph Pad Prism [version 8.00, Graph Pad Software, La Jolla, CA, USA], with *p* < 0.05 considered as statistically significant.

## 3. Results and Discussion

### 3.1. Chemical Characteristic

#### 3.1.1. Attenuated Total Reflection Fourier-Transform Infrared (ATR-FTIR) Spectroscopy

The ATR-FTIR spectroscopy analysis was carried out to investigate the structural changes of bioscaffolds based on PLA, PVA, and different salts before and after thermal treatment at the molecular level. The FT-IR spectra of bioscaffolds are shown in Figure 2A. The PLA-PVA bioscaffolds show some characteristic peaks identified by the strong infrared absorption band in the region of 1650–1754 cm^−1^, which corresponds to the stretching vibration of the carbon-oxygen double bond (C=O). The band at 1187 cm^−1^ is assigned to the stretching vibration of the carbon-oxygen bond (C-O). The two peaks around 1448 and 1373 cm^−1^ correspond to the methyl groups of the PLA-PVA bioscaffolds. Moreover, the high-intensity peak that is positioned at 3399 cm^−1^ corresponds to the stretching vibration of the oxygen-hydrogen bond (O-H) [12,42,43]. Figure 2A, illustrates that the intensity and position of the absorption peak of the hydroxyl group changed with the addition of salts and thermal treatment. Addition of KMnO_4_ to the bioscaffolds caused the disappearance of the O-H peak in the FT-IR spectra, which indicates the oxidation of PVA to polyvinyl ketone (PVK). The formation of corresponding ketone is due to the fact that KMnO_4_ is a strong oxidizing agent [44,45,46].

In the case of the addition of NaCl, the hydroxyl peak in PLA-PVA/NaCl bioscaffold is weaker than it is in PLA-PVA bioscaffold and its position shifted slightly towards a higher frequency (3430 cm^−1^). This can be attributed to a higher degree of hydrogen bonding in the PLA-PVA bioscaffold, because hydrogen bonding is disrupted by the addition of NaCl [47]. The thermal treated PLA-PVA and PLA-PVA/NaCl bioscaffolds show that the hydroxyl peak has a lower intensity in comparison with the non-thermal treated bioscaffolds [48]. In case of the thermal treated PLA-PVA/KMnO_4_ bioscaffold, the O-H peak in the FT-IR spectra appear and it shifted to a higher wavelength from (3353 cm^−1^). This can be explained by the partial formation of the carboxylic acid group due to the elevated temperature in the presence of KMnO_4_ [49,50]. Moreover, oxidation process led to an increase in the carbon-oxygen double bond (C=O) peak as can be seen in Figure 2B [51,52].

#### 3.1.2. Thermal Properties

DSC analysis of the porous bioscaffolds was carried out to study the effects of the thermal treatment and salts on the thermal behavior of the bioscaffolds. The correlated thermal properties of the bioscaffolds are summarized in Table 1. As expected, crystallinity (χ_c_) increased after the thermal treatment of the bioscaffolds [31]. The T_g_ of the bioscaffolds did not demonstrate any significant changes after the thermal treatment, as shown in Figure 3. Furthermore, T_m,_ and T_c_ values were not affected by the thermal treatment and were in agreement with the published literature values [31,53]. Increase in the concentration of NaCl caused significant reduction in crystal_l_inity, which could be attributed to the fact that high NaCl content impedes PLA chain mobility and thereby prohibits crystallization [54]. According to the literature, the oxidation of PLA and PVA by KMnO_4_ caused a decrease in crystallinity [55,56]. Moreover, the addition of salts induced a decrease in T_c,_ an increase in T_m_ (consistent with the results given in [57]), and showed no difference in the T_g_ values.

#### 3.1.3. Measurement of Porosity

The study of the porosity of wound dressings is a very important factor as it affects the absorption capacity of exudates from the wound, which can reduce the probability of infection [58]. As shown in Table 2, all the bioscaffolds showed a porosity ranging between 68.1 ± 1.2 and 94.1 ± 1.5% with statistical significance (*p* < 0.05). These results indicate that the porosity was increased with the addition of the salts. A higher salt concentration led to an increase in the porous structure, resulting in an irregular and spongier shape [27]. However, the total porosity of the bioscaffolds slightly decreased after the thermal treatment [29,30,59]. The graph clearly shows that non-thermal treated bioscaffolds with the highest concentration (10 g/L) of NaCl possessed the highest porosity (94%), whereas there was a reduction of porosity (92%) for thermal treated bioscaffolds with the same concentration of NaCl. Addition of KMnO_4_ to the bioscaffolds also followed the same trend as NaCl in terms of porosity. It is worth noting that the thermal treated bioscaffolds without salt (neat bioscaffolds) showed the lowest levels of porosity (68%).

#### 3.1.4. Morphology of the Porous Bioscaffolds

The optical photographs PLA-PVA porous bioscaffolds is shown in Figure 4. PLA-PVA has a white appearance with a smooth surface. The final dimension of produced samples is a circle with a diameter of 40 mm and a thickness of 5 mm. Scanning electron microscopy (SEM) was used to investigate the effects of the addition of salts and thermal treatment on the morphology of porous bioscaffolds. As can be observed from Figure 5, the PLA-PVA bioscaffold shows a disordered, interconnected pore-like structure with a rough surface. However, high resolution SEM analysis (Figure 6a shows that the neat PLA-PVA bioscaffold has fracture-like characteristics with almost no holes or pores. The addition of 10 g/L NaCl to the bioscaffolds led to the formation of more porous structures with interconnected pores with varying pore sizes in the range of 0.2–7 µm (Figure 6b). As can be seen from Figure 6c, the addition of 0.4 mg/L KMnO_4_ to the bioscaffolds led to the formation of a porous structure in the range of 0.4–4 µm. However, comparing Figure 6b,c leads to two main observations: (1) The addition of 10 g/L NaCl to the bioscaffolds led to the formation of more porous structures compared to the addition of 0.4 mg/L KMnO_4_ to the bioscaffolds; and (2) the addition of different salts led to the formation of different shapes and sizes of the pores. It is likely that the size and shape of the pores differ due to the oxidation of PVA by KMnO_4_ (discussed in FT-IR analysis section) in the case of PLA-PVA/KMnO_4_ bioscaffolds. Pore formation of the bioscaffolds containing salts can be attributed to its osmotic properties in the aqueous phase, which has already been demonstrated by other authors [27,60]. The bioscaffolds that have been subjected to thermal treatment present smoother surfaces with fewer fractures and less porosity (Figure 6). These results were consistent with other studies [30,59].

#### 3.1.5. Swelling Test

The degree of swelling is mainly dependent on the porosity and hydrophilicity of the bioscaffolds [35,61]. This property of the bioscaffolds plays an important role in the acceleration of wound healing as it absorbs exudates and fluids secreted from the wound and provides a moist environment for the wound area [3,8]. Figure 7 shows the swelling behavior of the bioscaffolds. It was observed that increasing porosity led to an increase in the swelling degree. The highest swelling degree was obtained by the non-thermal treated PLA-PVA/NaCl bioscaffolds. This can be attributed to the high porosity percentage of PLA-PVA/NaCl compared to other bioscaffolds (Table 2). The lowest swelling degree was obtained from thermal treated neat PLA-PVA bioscaffolds due to its lowest porosity and relatively high crystallinity [62]. The higher swelling degree of thermal treated PLA-PVA/KMnO_4_ bioscaffolds compared to non-thermal treated PLA-PVA/KMnO_4_ bioscaffolds may be attributed to the formation of the carboxylic acid group by oxidation of PVK in the presence of KMnO_4_ at an elevated temperature (80 °C). Therefore, the higher swelling degree of thermal treated PLA-PVA/KMnO_4_ bioscaffolds is attributed to the higher polarity and hydrophilic character of the carboxylic acid [57,63,64]. Solvent residue analyses were performed and values of chloroform approximately 8 ppm were found.

#### 3.1.6. Water Solubility

The water solubility of a polymer is a key factor in wound dressing applications, as the rate of degradation or hydrolysis takes place simultaneously with the wound healing process. If the degradation of the wound dressing occurs before the completion of the wound healing process, the wound dressing will need to be applied on the patient several times. This will not only cause discomfort but will also impose extra costs on the patient [65]. The water solubility assessment was performed by calculating the weight loss of the bioscaffolds in DI water after 24 h using Equation (4). Water solubility of the bioscaffolds ranges from 2% for thermal treated PLA-PVA and 10% for non-thermal treated PLA-PVA/NaCl as indicated in Table 3. These results reveal that water solubility increases by increasing the amount of the salts in the bioscaffolds. This increase in water solubility can be attributed to an increase in the porosity of the bioscaffolds, which is a result of the addition of salts. More porous structures allow and retain a higher number of water molecules in their structure [37,65]. It is worth noting that all the bioscaffolds kept their initial shape even after 24 h. Notably, the thermal treatment of the bioscaffolds did not significantly affect the values of water solubility.

#### 3.1.7. Water Vapor Transmission Rate (WVTR)

Water vapor transmission rate (WVTR) is the measurement of the amount of water lost through the dressing material [58]. An ideal wound dressing material should protect the wound from dehydration, which will occur due to high WVTR. It should also protect the wound from the accumulation of exudates and the risk of bacterial growth caused by low WVTR [66,67]. To maintain a moist environment for better wound healing the optimal range of WVTR for wound dressing material is 2000–2500 (g/m^2^·day) [39,68]. As shown in Table 4, the measured value of WVTR of the bioscaffolds were in the range of 2115–2287 g/m^2^·day (*p* < 0.05). As previously mentioned, the addition of salts increased the porosity of the bioscaffolds. This increase in the porosity is the main reason for the observed increase in the values of the WVTR in PLA-PVA/NaCl and PLA-PVA/KMnO_4_ bioscaffolds. The thermal treatment of the bioscaffolds did not affect the values of WVTR. The obtained WVTR results demonstrate that the bioscaffolds are suitable for wound dressing applications [39,68].

### 3.2. In Vitro Drug Release Studies

An ideal antimicrobial wound dressing should sustain a long period of controlled drug release in order to accelerate the healing process and to avoid frequent changing of the dressing [49]. Gentamicin sulfate as an antibiotic agent was loaded into the PLA-PVA bioscaffolds. The effect of the addition of different types of salts and thermal treatment on entrapment efficiency (EE), loading capacity (LC), and in vitro were studied. Tang et al. reported that the EE of drugs in the surface liquid spraying method is higher than the EE of drugs in the traditional emulsion solvent evaporation method [18]. Therefore, the liquid spraying method was used to obtain a higher EE. As shown in Table 5, the surface liquid spraying method resulted in a high entrapment efficiency of the drug (90.11%). Furthermore, the addition of salts increased the EE. This could be attributed to the changing of the aqueous solubility of the organic solvent by salt [24,69]. This could also be explained by increasing the porosity of the bioscaffolds due to the addition of salts as mentioned in the porosity measurement section [27,70]. While the thermal treatment did not significantly impact the EE, this was not the case for the PLA-PVA/NaCl. This could be attributed to the reduced porosity of PLA-PVA/NaCL due to the thermal treatment. Table 5 demonstrates that the addition of salts and thermal treatment did not affect LC (%), which can be attributed to the strong dependency of LC on the polymer weight ratio in accordance with Equation (7).

The addition of salts to the polymer has a crucial effect on the initial burst release and the porosity of the bioscaffolds. The initial burst release and the porosity of the bioscaffolds vary depending on the salt concentration [27]. The in vitro release profiles of antibiotics from the wound dressings are displayed in Figure 8. PLA-PVA/NaCl bioscaffolds exhibited the highest initial burst release due to the highest salt concentration and porosity. The cumulative drug release was around 82%. The burst release rate during the first 24 h in PLA-PVA/NaCl bioscaffolds can be attributed to the fact that the aqueous environment washed all the drugs from the surface, and other nearby drugs were removed through the pores of the polymer matrix [70,71]. In comparison with PLA-PVA/NaCl bioscaffolds, thermal treated PLA-PVA/NaCl bioscaffolds showed an initial burst release of drugs during the first 6 h of around 11%. This clearly showed a reducing initial burst release followed by a gradual release at a decreasing rate over time, with around 50% release of the drug during 14 days. These results are consistent with the results of other groups where thermal treatment was employed as a tool for prolonging the release of the drug. Moreover, thermal treatment of polymer at temperatures above T_g_ reduced the drug release rate [28,72]. This can be attributed to the fact that thermal treatment increases the crystallinity of the polymer, where crystalline domains function as a physical barrier, leading to slower diffusion of the drug [31]. As a result, thermal treatment of the PLA-PVA/NaCl bioscaffolds cause the sustained release of GS. However, due to the heating of the bioscaffolds above T_g_ (80 °C), the drug release rate was reduced.

As shown in Figure 8, for PLA-PVA/KMnO_4_ bioscaffolds, the initial burst release of drugs during the first 24 h was only approximately 20%, followed by a gradual and constant release of GS over 14 days. The cumulative drug release was 33%. However, thermal treated PLA-PVA/KMnO_4_ bioscaffolds showed an initial burst release of around 12% during the first 6 h, followed by a fast sustained release profile around 61% over 14 days. As can be seen in the figure, the cumulative drug release rate of heat treated PLA-PVA/KMnO_4_ bioscaffolds had a higher release rate in comparison with PLA-PVA/KMnO_4_ bioscaffolds. This could be explained by the formation of PVK as the result of the interaction between KMnO_4_ and PVA. The thermal treating of PLA-PVA/KMnO_4_ bioscaffolds caused partial oxidation of the formed PVK by KMnO_4_, and resulted in the formation of carboxylic acid groups. Carboxylic acids have a higher polarity and hydrophilic character in comparison with ketones (PVK) [57,63,64]. Therefore, thermally treated PLA-PVA/KMnO_4_ bioscaffolds have a relatively higher hydrophilicity as compared to non-thermal treated PLA-PVA/KMnO_4_ bioscaffolds. This higher hydrophilicity causes PBS to permeate more freely into thermal treated PLA-PVA/KMnO_4_ bioscaffolds than it can permeate into the PLA-PVA/KMnO_4_ bioscaffolds. Hence, although thermal treatment of the PLA-PVA/KMnO_4_ bioscaffolds led to a decrease in porosity, its higher relative hydrophilic character caused the higher cumulative release rate.

For the neat PLA-PVA bioscaffolds, the initial burst release occurred during the first 6 h followed by a slow and gradual release at around 58% over 14 days (Figure 8). The thermally treated neat PLA-PVA bioscaffolds exhibited the initial burst release in the first 6 h at around 14% while the cumulative drug release was 20%. This means that the thermally treated neat PLA-PVA bioscaffolds could not release the drugs and kept the drug inside the bioscaffolds. This can be attributed to the less porous structure and relatively higher crystallinity of the bioscaffolds [73].

## 4. Conclusions

Polymeric porous bioscaffolds of PLA-PVA were prepared in this study by a slightly modified form of surface liquid spraying method. The effects of the addition of different salts (NaCl and KMnO_4_) and thermal treatment (80 °C for 2 min) on the bioscaffolds were investigated. The SEM results indicated that prepared bioscaffolds had interconnected porous structures and the addition of salts considerably enhanced the porosity of the bioscaffolds. Moreover, the swelling degree and water solubility of bioscaffolds were increased due to the increase in porosity. The in vitro release of gentamicin sulfate was studied and it was shown that a higher entrapment efficiency and initial burst release was achieved by the addition of salt to the aqueous phase. Additionally, the thermal treatment of the polymer above T_g_ reduced the initial burst release and prolonged the release of the drug. Finally, it worth noting that the procedure suggested in this study to prepare bioscaffolds is cost-efficient and non-toxic, since all the solvents can be easily and completely removed. Therefore, the novel PLA-PVA bioscaffolds developed in this work could be a potential candidate for wound dressing applications in the future.

## Figures and Tables

**Figure 1 polymers-13-00921-f001:**
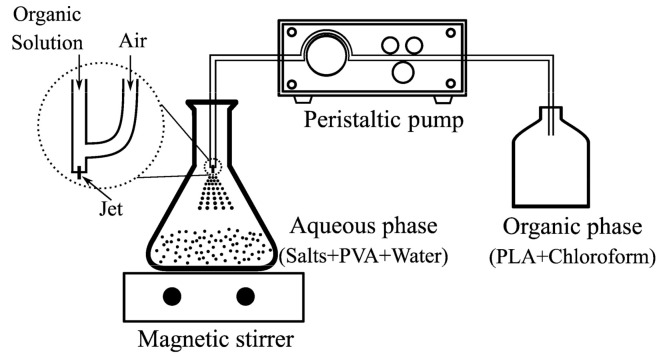
Scheme of the surface liquid spraying process.

**Figure 2 polymers-13-00921-f002:**
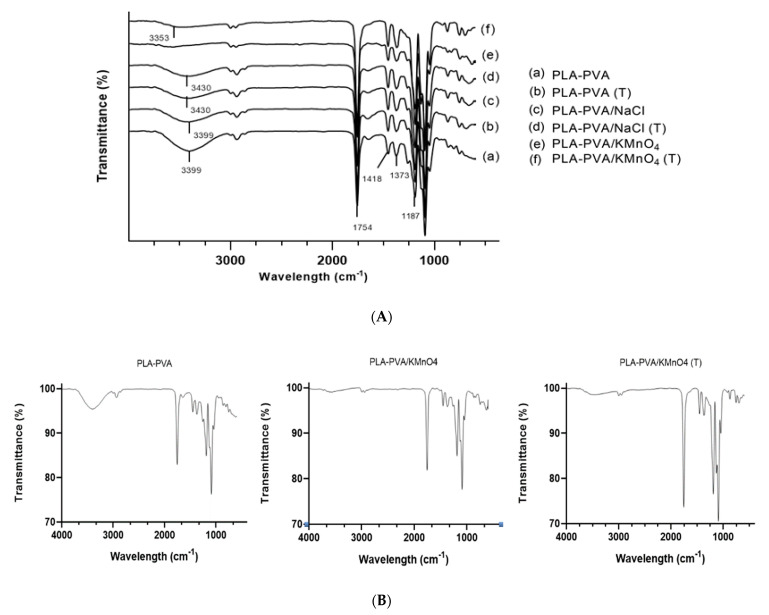
(**A**). FT-IR spectra of PLA-PVA (a), PLA-PVA (T) (b), PLA-PVA/KMnO_4_ (c), PLA-PVA/KMnO_4_ (T) (d), PLA-PV/NaCl (e), PLA-PVA/NaCl (T) (f) bioscaffolds. (**B**). FT-IR spectra of PLA-PVA, PLA-PVA/KMnO_4_, and PLA-PVA/KMnO_4_ (T) of bioscaffolds.

**Figure 3 polymers-13-00921-f003:**
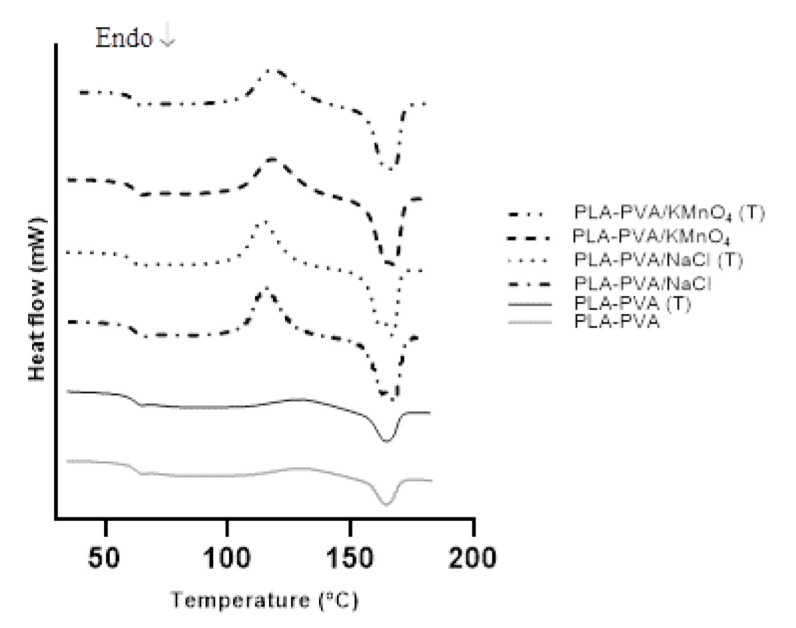
DSC curves of porous PLA bioscaffolds.

**Figure 4 polymers-13-00921-f004:**
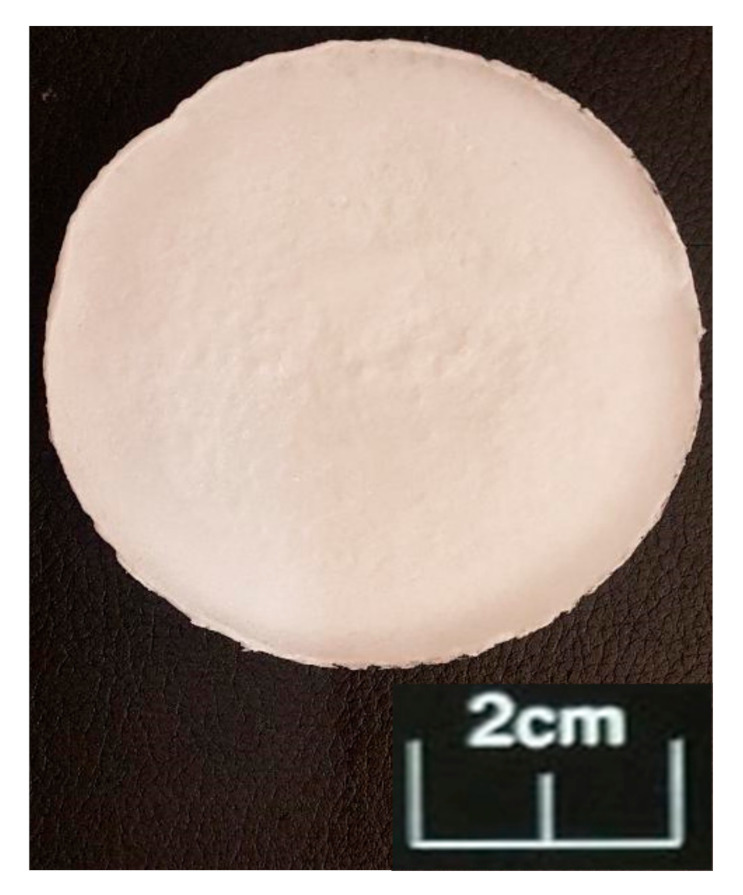
Photographic appearance of the PLA-PVA porous bioscaffold.

**Figure 5 polymers-13-00921-f005:**
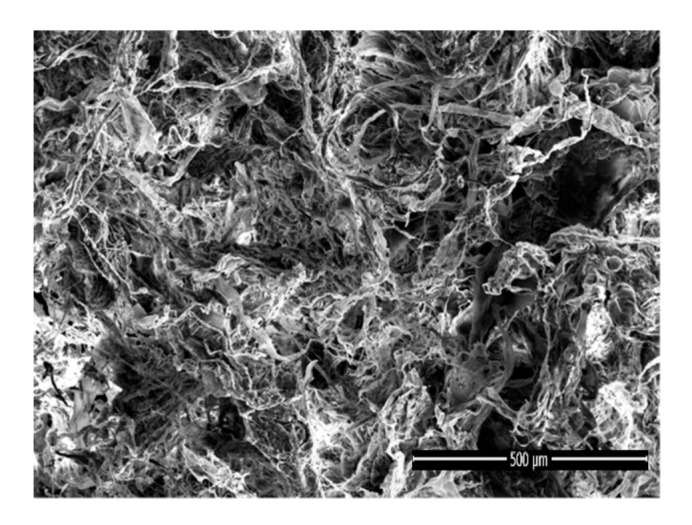
SEM image of PLA-PVA porous bioscaffold.

**Figure 6 polymers-13-00921-f006:**
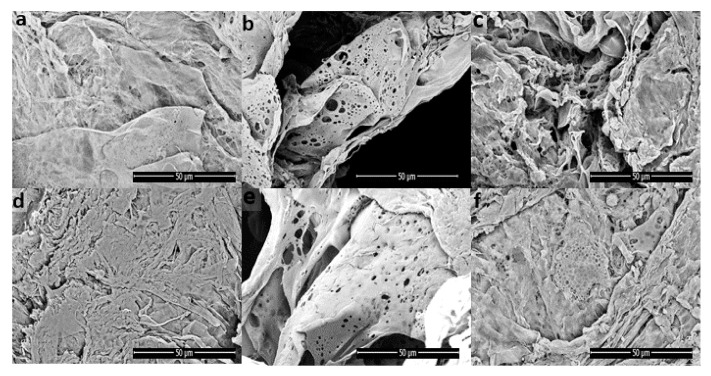
SEM images of the porous bioscaffolds: (**a**) PLA-PVA, (**b**) PLA-PVA/NaCl, (**c**) PLA-PVA/KMnO_4_, (**d**) PLA-PVA (T), (**e**) PLA-PVA/NaCl (T), and (**f**) PLA-PVA/KMnO_4_ (T).

**Figure 7 polymers-13-00921-f007:**
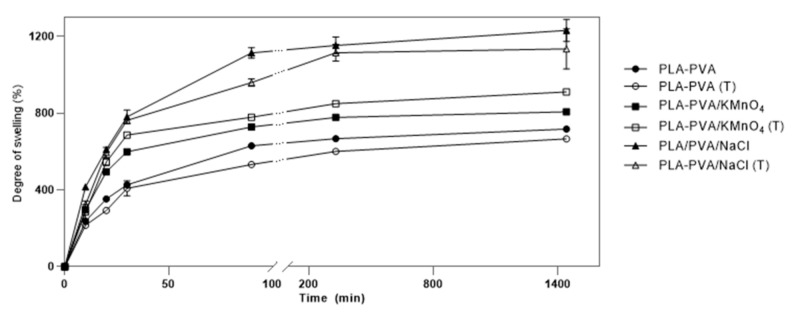
Swelling studies of the porous bioscaffolds in phosphate-buffered saline (PBS) with pH 7.4 at 37 °C.

**Figure 8 polymers-13-00921-f008:**
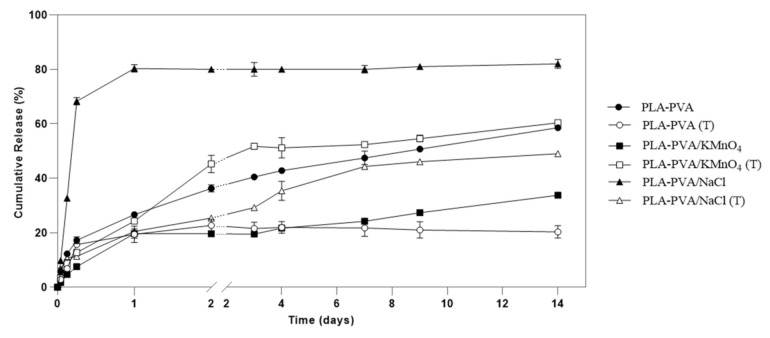
In vitro release profiles of porous bioscaffolds loaded gentamicin sulfate (GS) in pH 7.4 at 37 °C.

**Table 1 polymers-13-00921-t001:** Selected material-related properties of the bioscaffolds before and after thermal treatment.

	Untreated			After Thermal Treatment		
Sample/Property	PLA-PVA	PLA-PVA/KMnO_4_	PLA-PVA/NaCl	PLA-PVA	PLA-PVA/KMnO_4_	PLA-PVA/NaCl
χ_c_	5.1	4.9	0.7	8.8	5.3	3.0
T_g_	61.9	62.0	62.0	61.1	61.7	61.2
T_c_	129.9	118.3	115.4	130.4	118.1	114.4
T_m_	164.3	167.4	168.0	164.6	167.5	167.0

**Table 2 polymers-13-00921-t002:** Porosity measurements of porous bioscaffolds after 24 h of immersion in ethanol at room temperature.

	Untreated			After Thermal Treatment		
Sample	PLA-PVA	PLA-PVA/KMnO_4_	PLA-PVA/NaCl	PLA-PVA	PLA-PVA/KMnO_4_	PLA-PVA/NaCl
Porosity	69.87 ± 2.1	78.77 ± 4.2	94.1 ± 1.5	68.1 ± 1.2	75 ± 0.94	92.86 ± 0.54

*p* value < 0.05.

**Table 3 polymers-13-00921-t003:** Water solubility measurements of the porous bioscaffolds in DI water after 24 h at 37 °C.

	Untreated			After Thermal Treatment		
Sample	PLA-PVA	PLA-PVA/KMnO_4_	PLA-PVA/NaCl	PLA-PVA	PLA-PVA/KMnO_4_	PLA-PVA/NaCl
Water solubility	4.09 ± 0.00	6.50 ± 0.20	10.87 ± 0.18	2.67 ± 0.25	7.36 ± 0.18	9.41 ± 0.50

**Table 4 polymers-13-00921-t004:** Water vapor transmission rate of the porous bioscaffolds.

	Untreated			After Thermal Treatment		
Sample	PLA-PVA	PLA-PVA/KMnO_4_	PLA-PVA/NaCl	PLA-PVA	PLA-PVA/KMnO_4_	PLA-PVA/NaCl
WVTR	2146.01 ± 19.48	2214.16 ± 20.94	2287.61 ± 25.31	2077 ± 18.65	2207.08 ± 24.45	2259.3 ± 21.35

*p* value < 0.05.

**Table 5 polymers-13-00921-t005:** Encapsulation efficiency (EE) and loading capacity (LC) of the porous bioscaffolds before and after thermal treatment.

Properties	Untreated Treatment			After Thermal Treatment		
	PLA-PVA	PLA-PVA/KMnO_4_	PLA-PVA/NaCl	PLA-PVA	PLA-PVA/KMnO_4_	PLA-PVA/NaCl
EE (%)	90.11 ± 0.21	92.08 ± 0.06	97.57 ± 0.03	92.38 ± 0.49	93.52 ± 0.46	86.7 ± 0.4
LC (%)	4.5 ± 0.01	4.6 ± 0.003	4.8 ± 0.001	4.6 ± 0.02	4.7 ± 0.02	4.32 ± 0.02

*p* value < 0.05.

## Data Availability

Not applicable.

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
