# Peer review of "Polylactide/Polyvinylalcohol-Based Porous Bioscaffold Loaded with Gentamicin for Wound Dressing Applications"

_polymers, 2021, doi:10.3390/polym13060921_

Round 1

Reviewer 1 Report

The manuscript “Polylactide-polyvinylalcohol-based porous systems loaded with gentamicin for wound dressing applications” deals with the preparation of gentamicin sulfate loaded polylactide-polyvinyl alcohol by spray drying. The work is interesting and several characterizations have been performed. However, the publication is recommended after some revisions.

In particular:

- Abstract. Please, add quantitative results to this paragraph.

- Introduction. The state of the art can be enlarged, adding other studies in the field, with the aim of highlighting the novelty of this work. See, for instance, Cardea et al., 3D PLLA/Ibuprofen composite scaffolds obtained by a supercritical fluids assisted process, Journal of Materials Science: Materials in Medicine, 2014, 25, pp. 989-998; etc...

- Materials. Please, add molecular weight and other characteristics to the materials used for the experimentation.

- Results and Discussion. Since the produced materials could be applied to the medical field, a solvent residue analysis should be performed.

Authors stated that “However due to the heating of the mats above Tg (80°C) the drug release rate was reduced”. But, a temperature of 80 °C could have a negative effect on the drug? Please, discuss this aspect.

Which is the final dimension of the produced mats?

- English should be improved.

Author Response

Dear reviewer,
Please find point by point and manuscript in the attachment.
Regards,
Maliheh Amini Moghaddam

Reviewer 2 Report

Please find the pdf doc.

Author Response

(The authors gave the same response as above.)

Reviewer 3 Report

Dear Authors,

In this manuscript, a PLA-PVA polymeric porous mats were obtained and characterised by robust physicochemical methods. As presented, this content seemed acceptable for publication.

Author Response

Dear reviewer,
Thank you for the comment.

Best Regards,
Maliheh Amini Moghaddam

Round 2

Reviewer 1 Report

The authors performed all the modifications proposed by the Reviewer.

Minor observations:

  • some typing errors are present. Please, check and correct them;
  • please, change ml with mL.

Author Response

Dear reviewer,
Thank you for the comment. The manuscript was updated to reflect this.
Best Regards,
Maliheh Amini Moghaddam

Reviewer 2 Report

1) The value in abstract should be come with ± for scientific writing.

2) Figure 1 should be improved..low quality/resolution.

3) Grh pad prism info should be in bracket (version manufacturer, country)

4) Figure 4 is not clear enough..please change and set high resolution

5) Scale bar for Figure 5 is not clear..change/revise

6) Figure 8..SI unit should be come with gap....

7) the term in vitro and in vivo are universal..no need "-"...please change throughout the manuscript

Author Response

Dear reviewer,
Thank you for the comment. Please find the point-by-point draft in the attachment.

Best Regards,
Maliheh Amini Moghaddam
